# The DEAD-box ATPase Dbp10/DDX54 initiates peptidyl transferase center formation during 60S ribosome biogenesis

Victor E. Cruz [1,2], Christine S. Weirich [1], Nagesh Peddada [1,3] & Jan P. Erzberger [1] ✉

DEAD-box ATPases play crucial roles in guiding rRNA restructuring events during the biogenesis of large (60S) ribosomal subunits, but their precise molecular functions are currently unknown. In this study, we present cryo-EM reconstructions of nucleolar pre-60S intermediates that reveal an unexpected, alternate secondary structure within the nascent peptidyl-transferase-center (PTC). Our analysis of three sequential nucleolar pre-60S intermediates reveals that the DEAD-box ATPase Dbp10/DDX54 remodels this alternate base pairing and enables the formation of the rRNA junction that anchors the mature form of the universally conserved PTC A-loop. Post-catalysis, Dbp10 captures rRNA helix H61, initiating the concerted exchange of biogenesis factors during late nucleolar 60S maturation. Our findings show that Dbp10 activity is essential for the formation of the ribosome active site and reveal how this function is integrated with subsequent assembly steps to drive the biogenesis of the large ribosomal subunit.

Accurate assembly of large (60S) and small (40S) ribosomal subunits requires an elaborate, multicompartmental biogenesis pathway[1,2]. During this maturation process, primary rRNA transcripts undergo co-transcriptional modifications and nucleolytic processing before folding into a compact tertiary structure stabilized by interactions with ribosomal proteins (RPs). The initial, spontaneous folding of primary rRNA transcripts favors the formation of local stem loops, which must then assemble into correct helical junctions using long-range secondary and tertiary interactions to form larger rRNA domains[3,4] (Supplementary Fig. 1a). RPs guide and accelerate this process, stabilizing unique tertiary structures to define the ribosome assembly pathway. In eukaryotes, this process is further facilitated by more than 200 ribosome biogenesis factors (RBFs)[5,6] that transiently engage pre-ribosomes in a coordinated, hierarchical manner to enable the high rates of ribosome production required for cellular growth and division[7].

One crucial rRNA element of the 60S is the universally conserved peptidyl transferase center (PTC), which brings together A- and P-site tRNAs and catalyzes peptide bond formation. Within the PTC lies the A-loop, which caps helix H92 and orients the CCA tail of the A-site tRNA, ensuring the proper orientation of the acceptor amino acid within the PTC[8]. The position of the A-loop within the PTC is defined by an intricate three-way junction that anchors rRNA helices H90, H91 and H92 and is only peripherally bound by RPs[9]. To date, there is no structural information on how these tertiary rRNA interactions form during 60S assembly.

PTC assembly and/or quality control has been proposed to require the activity of DEAD-box ATPases, ubiquitous RNA-directed enzymes with wide-ranging functions throughout RNA biology[10–14]. In prokaryotes, the DEAD box protein DbpA directly associates with H92 via a C-terminal RNA-binding domain[15,16]. While there is no eukaryotic homolog of DbpA, Dbp10/DDX54, another DEAD-box ATPase, has been linked to rRNA helices H90-H91-H92 through CRAC analysis[17]. More recently, cross-linking mass spectrometry and structural studies have placed Dbp10/DDX54 on the nascent subunit interface, flanked by the RBFs Nop2, Nog1 and Nsa2[18–20], but these studies have not

[1]Department of Biophysics, UT Southwestern Medical Center - ND10.124B, 5323 Harry Hines Blvd., Dallas, TX 75390, USA. [2]Present address: O'Donnell Brain Institute/CAND, UT Southwestern Medical Center, 5323 Harry Hines Blvd., Dallas, TX 75390, USA. [3]Present address: Center for the Genetics of Host Defense, UT Southwestern Medical Center, 5323 Harry Hines Blvd., Dallas, TX 75390, USA. ✉e-mail: jan.erzberger@utsouthwestern.edu

defined the catalytic function of Dbp10/DDX54 or revealed how the early PTC is formed.

In this study, we set out to investigate the possible role of Dbp10 in PTC formation during nucleolar pre-60S assembly. Using a slow-hydrolysis mutant, we pinpoint the precise timing of Dbp10 catalytic function during 60S maturation and present three cryo-electron microscopy (cryo-EM) reconstructions of pre-catalytic, catalytic and post-catalytic pre-60S states that establish the function of Dbp10 in PTC formation. The pre-catalytic structure reveals an unexpected, alternate secondary structure of helix H92 prior to Dbp10 engagement. Unwinding and remodeling of this rRNA structure by Dbp10 forms the A-loop and facilitates the assembly of the H90/H91/H92 three-way junction at the core of the PTC. Finally, the post-catalytic Dbp10-pre60S complex communicates these structural changes in the PTC to other regions of the pre-60S, coupling the formation of the ribosome active site to the compositional re-structuring of the pre-60S particle.

## Results

### Genetic and structural characterization of Dbp10 function

Because Dbp10 has been purified as part of several pre-60S particles during nucleolar 60S maturation, we first set out to define the precise timing of Dbp10 catalytic rRNA engagement. The active site architecture at the interface of the two RecA domains (D1 and D2) that form the enzymatic core of DEAD-box ATPases is highly conserved, allowing us to use insights from previous studies[21–23] to generate Dbp10 mutants that fail to bind ATP and substrate (Dbp10[K187A], Dbp10[K187R], and Dbp10[D288A]) or mutants that can bind ATP and ssRNA, but are deficient in hydrolysis (Dbp10[E289A] and Dbp10[R522V]) (Fig. 1a). Genetically, these slow-hydrolysis mutants are predicted to become kinetically trapped on the pre-60S substrate and therefore display a dominant negative phenotype in vivo.

In a *DBP10* shuffle strain, *dbp10[K187A]*, *dbp10[K187R]* and *dbp10[D288A]* all fail to rescue the lethality of a *dbp10Δ* allele, while mutants *dbp10[E289A]* and *dbp10[R522V]* display a slow growth phenotype (Fig. 1b). Consistent with our hypothesis, estradiol-inducible overexpression of *dbp10[R522V]* results in a dominant negative phenotype (Fig. 1c), while no effect is observed upon overexpression of variants mutated in motifs I (*dbp10[K187A]*) or II (*dbp10[D288A]*) (Fig. 1c), likely because these mutants never engage the pre-60S. The growth defect of the *dbp10[R522V]* strain is due to a 60S maturation defect, as determined by polysome profiling of *dbp10[R522V]* extracts (Fig. 1d). Monitoring of ribosome biogenesis in vivo using GFP-tagged large (uL23-GFP) or small (uS7-GFP) subunits shows that the observed decrease in mature 60S subunits is due to nuclear/nucleolar accumulation of pre-60S subunits (Fig. 1e). To identify the pre-60S substrate for Dbp10, we overexpressed *DBP10* or *dbp10[R522V]* and purified pre-60S intermediates with split tags on Dbp10 and the RBF Brx1, which is present on pre-60S intermediates before and after Dbp10 binding. Semi-quantitative mass-spectrometry analysis of these purified intermediates reveals that, compared with overexpression of *DBP10*, overexpression of *dbp10[R522V]* leads to an increase in the abundance of Rrp14, Rrp15, and Ssf1 and a decrease in the levels of Noc2, Noc3, Spb1, and Nug1 (Fig. 1f). Our results are broadly consistent with a study of Dbp10 mutants published while this manuscript was under review[24]. This factor exchange also coincides with the docking of rRNA domain III to the pre-60S core, indicating that Dbp10 remains bound during this major assembly event (Fig. 2a). Thus Dbp10, like the DEAD-box ATPases Has1 and Spb4, remains bound to the pre-60S post-catalytically, a state recently captured in cryo-EM reconstructions of human and *C. thermophilum* pre-60S intermediates[19,20]. These studies also show that another RBF exchange, the removal of the Nsa1/Mak16/Rpf1/Rrs1 module by Rix7, appears to occur independently from Dbp10 function.

We next set out to isolate pre-60S intermediates representing pre-catalytic, catalytic and post-catalytic states of Dbp10, using distinct genetic and biochemical trapping strategies (Fig. 2a and Supplementary Fig. 1b, c): Purification of pre-60S particles from a strain overexpressing *dbp10[R522V]* with split-tags on Dbp10[R522V] and Ssf1 yielded purified particles with stoichiometric amounts of Dbp10. However, cryo-EM reconstructions of this sample yielded a reconstruction that lacked any density for Dbp10, likely because it dissociated during grid preparation. A skip align 3D-classification focused on the region predicted by XL-MS to represent the binding region of Dbp10[18] revealed a subset of particles with a distinct, previously uncharacterized structural feature. Because Dbp10[R522V] is catalytically impaired, we reasoned that the resulting 2.5 Å reconstruction might represent the pre-catalytic state. (Fig. 2b, Supplementary Figs. 2, 3). We confirmed this by reconstructing the structure of pre-60S particles from a strain overexpressing a slow-hydrolysis mutant of a different DEAD-box ATPase, Drs1(Cruz, et al., unpublished data), which binds Ssf1-containing pre-60S particles immediately before Dbp10[18,25,26]. Pre-60S particles isolated from this strain lack Dbp10 (Supplementary Fig. 1d) but retain the same PTC architecture as the reconstruction from the *dbp10[R522V]* overexpression strain (Cruz, et al., unpublished data and Supplementary Fig. 1e).

A 2.7 Å reconstruction of the catalytic Dbp10 intermediate was obtained from pre-60S intermediates purified from wild-type Dbp10 steady state cells using split affinity tags on Dbp10 and Ssf1, stabilized by the addition of the γ-phosphate mimic beryllium fluoride (BeF$_3$) and low concentrations of the bifunctional crosslinker DSS (Supplementary Figs. 1, 4 and 5). BeF$_3$ allows DEAD box ATPases to unwind dsRNA and subsequently stabilizes a DEAD-box-BeF$_3$-ssRNA (i.e. substrate bound) complex[23,27–29]. This structure is compositionally identical to the pre-catalytic state, except for the presence of Dbp10 (Fig. 2b, c and Supplementary Fig. 1b, c).

Finally, the post-catalytic state was obtained from a strain harboring a genomic 3′ auxin-inducible degron (AID) fused to the DEAD-box ATPase *SPB4*. Addition of 3-indole acetic acid (auxin) to these strains led to a significant enrichment of the Dbp10 post-catalytic state (Supplementary Fig. 1c), purified using split affinity tags on Dbp10 and Noc2. A single particle reconstruction from this sample yielded a 2.7 Å reconstruction (Fig. 2d, Supplementary Figs. 6 and 7). The molecular composition of these three intermediates is consistent with our mass-spectrometry analysis, confirming the concerted exchange of essential RBFs during late nucleolar assembly while Dbp10 remains engaged on the pre-60S: The pre-catalytic and catalytic structures are characterized by the presence of Rrp14, Rrp15, and Ssf1, while in the post-catalytic state, these factors are replaced by Noc2, Noc3 and Spb1 (Fig. 2 and Supplementary Fig. 1b, c).

### Structure of the immature PTC

Our reconstruction of the pre-60S prior to Dbp10 catalysis captures the PTC immediately after the initial docking of three of its constituent rRNA helices (H89, H91 and H93) to the pre-60S core (Fig. 3a). This initial step is mediated by a set of RBFs that individually stabilize these helices: Nsa2 and Nog1 engage both H89 and H91 (Fig. 3b, c), while Nip7 binds the H93 stem-loop (Fig. 3d). In previously characterized structures of pre-60S intermediates, the elements connecting these helices could not be resolved and were assumed to be too dynamic to be captured by cryo-EM[20,30]. Our improved reconstruction now shows that, unexpectedly, the H92 stem and the A-loop do not form co-transcriptionally, but instead fold into an alternate base-pairing arrangement (Fig. 3e, f). In our model, four A-loop nucleotides (G2922-C2925) are base paired with a complementary sequence (G2939-C2942) that is part of H90 in the mature ribosome (Fig. 3g). Rather than forming the compact RNA junction observed in the mature ribosome (Fig. 3h, i), the immature architecture of H90/H92 extends away from the pre-60S surface and is stabilized by the RP uL14, which engages two separate regions within the alternate H91/H90/H92 structure (Fig. 3g).

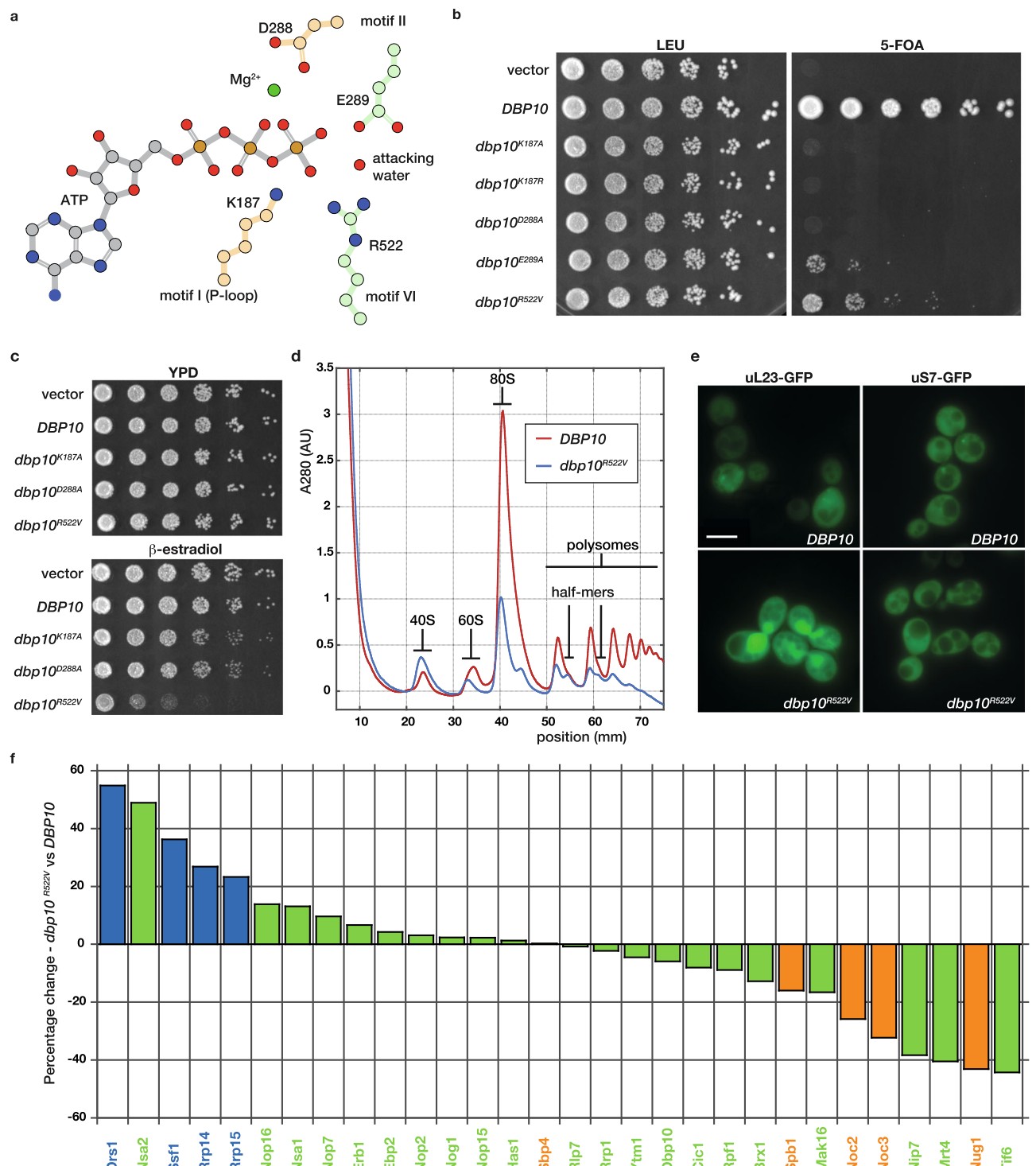

**Fig. 1 | Defining the timing of Dbp10 catalytic activity during 60S assembly.**
**a** Schematic of the predicted interactions between Dbp10 residues (peach/green) and ATP (gray) based on the conserved nucleotide binding pocket of DEAD-box ATPases. D288 and K187 (peach) stabilize the triphosphate and contribute to nucleotide binding. E289 and R522 (green) help orient the attacking water but do not directly stabilize ATP. **b** ATP binding is critical for Dbp10 function in a *DBP10* shuffle strain. Mutations predicted to have an ATP binding defect (*dbp10^{K187A/R}* or *dbp10^{D288A}*) cannot rescue lethality of *dbp10Δ*. Mutants predicted to bind but only slowly hydrolyze ATP (*dbp10^{E289A}* and *dbp10^{R522V}*) are viable but have a severe growth defect. **c** Overexpression of *dbp10^{R522V}* but not ATP binding mutants causes a dominant negative growth defect. **d** Polysome profiles from wild-type and *dbp10^{R522V}* strains. *dbp10^{R522V}* shows reduced levels of mature 60S and 80S as well as half-mer peaks characteristic of a 60S assembly defect. **e** In vivo distribution of GFP-tagged large (uL23-GFP) and small (uS7-GFP) ribosomal proteins. Images represent at least 3 independent experiments. Scale bar represents 5 μm. **f** Visualization of semi-quantitative mass-spectrometry data sorted by differences in abundance of nucleolar RBFs in *dbp10^{R522V}* vs *DBP10* strains. RBFs associated with earlier, Ssf1-containing intermediates (State C) are shown in light blue, RBFs associated with late-nucleolar Noc2/Noc3 particles (State E) are orange and RBFs present in both are green. Source data are provided as a Source Data file.

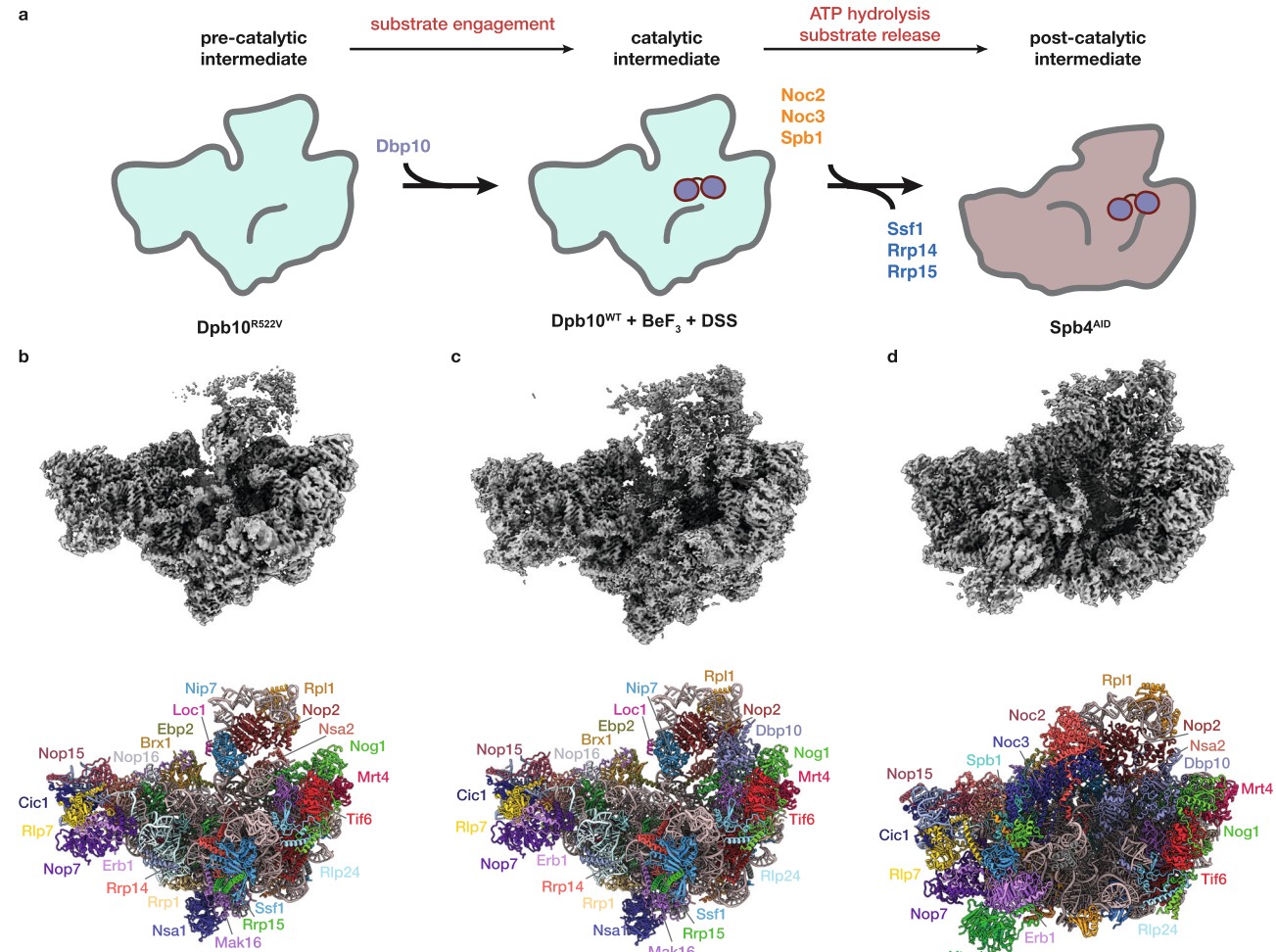

**Fig. 2 | Cryo-EM reconstructions of Dbp10 pre-catalytic, catalytic, and post-catalytic intermediates. a** Intermediate enrichment strategy. The pre-catalytic state was isolated using estradiol-induced overexpression of *dbp10^R522V^* followed by purification using affinity tags on Dbp10^R522V^ and Ssf1. The catalytic state was purified from steady state *DBP10* cell lysates treated with BeF3 using affinity tags on Dbp10 and Ssf1. The post-catalytic state was enriched by AID-degron mediated depletion of Spb4 and purification using affinity tags on Dbp10 and Noc2. Cryo-EM maps (top) and molecular models (bottom) of (**b**). pre-catalytic, (**c**) catalytic, and (**d**) post-catalytic pre-60S intermediates. RBFs present in each particle are colored and labeled.

Formation of a compact H90/H91/H92 junction is essential for A-loop stabilization of the aminoacylated CCA tail of the A-site tRNA (Fig. 3h). Therefore, the immature conformation we observe in the pre-catalysis intermediate must be restructured to form the functional PTC (Fig. 3i). Because RBF interactions with flanking rRNA helices (H89, H91 and H92) topologically restrict the pre-PTC scaffold, any RNA unwinding activity that locally destabilizes the alternate H90/H92 duplex will introduce a localized topological strain to facilitate the transition to the mature PTC secondary structure. Furthermore, since the number of base-pairs in the two conformations is comparable, the transition between the two secondary structures could be achieved merely by lowering the thermodynamic barrier between the two structures (Fig. 3f, h).

## Dbp10 unwinding activity destabilizes the immature A-loop

To stabilize the catalytic state of Dbp10, we purified steady-state intermediates from *DBP10* cells in the presence of beryllium fluoride (BeF3) to stabilize substrate-bound intermediates. The RBF composition of this intermediate is indistinguishable from the pre-catalysis state, except for extra density corresponding to the C-terminal portion of Dbp10 in the cleft between Nop2 and Nip7 on one side and Nsa2, Nog1 and uL14 on the other (Fig. 4a). We were able to build an atomic model for the D2 domain of Dbp10 as well as additional C-terminal elements into this map (Fig. 4b, c). The main 60S anchoring interaction

of Dbp10 is mediated by a series of helices within a conserved C-terminal extension (CTE) (Fig. 4b–d). The first three CTE helices wrap around the D2 domain in a triangular arrangement, mediating interactions with uL14, Nog1 and Nsa2, while an extended helical finger, composed of an additional pair of helices, points away from the D2 domain, making extensive interactions with Nog1, the N-terminal helix of Nsa2 and with rRNA helices H43 and H44 (Fig. 4d). Beyond the CTE, the C-terminal tail (CTT) of Dbp10 is entirely disordered in this reconstruction, except for a 16 amino acid segment (residues 781–796), that interacts with the RBF Nop2 (Fig. 4b–d). We tested the importance of these structural elements in vivo using a *DBP10* shuffle strain. Consistent with its pre-60S anchoring function, removal of the helical finger element (*dbp10^Δ667-731^*) results in a lethal phenotype (Fig. 4e). In contrast, alanine substitutions within the Nop2-interaction motif (*dbp10^ΔNop2int^*) results in a temperature-sensitive phenotype at 37 °C (Fig. 4f), suggesting that the Nop2 interaction is not required for initial binding, but may instead help stabilize the mobile RNA segment termed the L1 stalk (helices 75–78), especially at higher temperatures.

Previous structural and biochemical studies of DEAD-box dsRNA unwinding using the model protein Mss116 established different functions for the two core domains: The D1 domain binds ATP while the D2 domain binds one of the strands of the duplex substrate (Fig. 4g)[28,31,32]. Cooperative assembly of a bipartite interface then leads to strand unwinding and stabilization of a closed D1/D2 interface

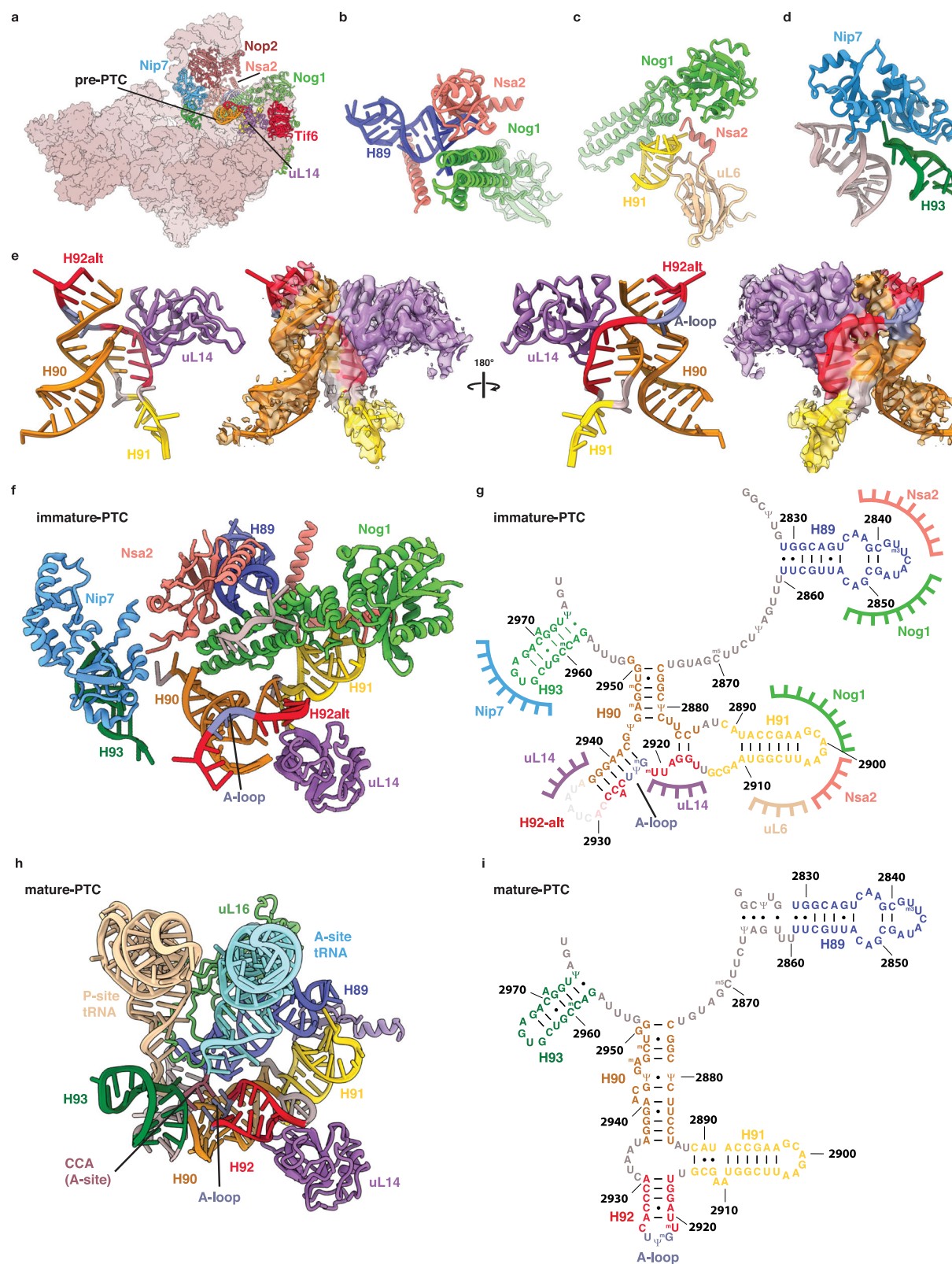

containing ATP and ssRNA[28,31,33]. We reasoned that if the alternate H90/H92 structure is the Dbp10 substrate, then it must be properly oriented for recognition by the D2 domain. To determine the relative positions of D2 in the catalytic state and H92alt in the pre-catalytic state, we aligned the two models using uL14 as a guide. Superposition of H92alt onto the catalytic model (Fig. 4h and Supplementary Movie 1) shows that the duplex region formed by A-loop nucleotides is in the proper

orientation to be engaged by the D2 domain, requiring only a ~4 Å rigid body swivel to place duplexed A-loop residues onto the dsRNA-recognition surface of the D2 domain of Dbp10 (Fig. 4i).

We observed no defined density for the D1 domain and the rRNA sequence spanning H90/H92 in our initial maps (Fig. 4j). We hypothesized that our BeF₃ trapping strategy may only have succeeded for a subset of particles; to isolate this subset we performed two rounds of

**Fig. 3 | Organization and structure of the Dbp10 substrate. a** Overall view of the pre-catalytic intermediate highlighting the RBFs and RPs (shown as cartoons and labeled) that organize the pre-catalytic PTC structure. **b** Cartoon depiction of the stabilization of H89 by Nog1 and Nsa2. **c** Cartoon depiction of stabilization of H91 by Nog1, Nsa2 and uL6. **d** Cartoon depiction of the stabilization of H93 by Nip7 and H74. **e** "Front" and "back" cartoon depiction of atomic model (left) and cryo-EM map (right - low-pass filtered to 2.75 Å), showing the alternate structure of H90/H92 and the position of uL14 in the pre-catalytic state. Nucleotides that form H90 in the mature ribosome are shown in orange, residues that will form H92 are shown in red and the A-loop is shown in blue. **f** Cartoon depiction of the PTC organization in the pre-catalytic intermediate, showing how the alternate H90/H92 region is flanked by immobilized stem loops. Nucleotides are colored and labeled according to their position in the mature rRNA secondary structure. **g** Secondary structure diagram of the immature PTC model shown in panel (**f**), highlighting the alternate base-pairing of H90/H92 as well as the contact points between RBFs, RPs and flanking rRNA helices. **h** Cartoon model of the mature *S.cerevisiae* PTC with A- (cream) and P-site (cyan) tRNAs (PDB-6TNU[9]), showing the compact, mature organization of helices H90, H91 and H92 and the A-loop A-site tRNA interaction. **i** Secondary structure diagram of the PTC rRNA in the mature 60S ribosomes as shown in panel (**h**).

skip-align local classification with masks centered on density observed in low-pass filtered maps near the expected position of the D1 domain (Supplementary Fig. 4). This strategy allowed us to identify a particle subset with additional density adjacent to the D2 domain (Fig. 4j). The local resolution of this density is ~6 Å, allowing us to manually dock a model for the Dbp10 D1 domain and rigid-body refine it against a map low-pass filtered to 6 Å (Fig. 4k).

A comparison of the resulting D1/D2 interface with a model of ssRNA-substrate bound Mss116 shows that the D1 model is in an orientation that closely matches the substrate-bound D1/D2 interface (Fig. 4j), although the rigid-body refined D1-D2 interface is not fully engaged and we observe no clear density for bound ssRNA. The absence of defined substrate ssRNA at the D1/D2 interface could be due either to the presence of multiple states representing intermediates between H92 alt and the mature PTC conformation or because of destabilization of the complex during grid preparation. In particular, because the DSS crosslinker added to stabilize our intermediate can readily form monolinks with lysine residues, we speculate that the Dbp10/ssRNA complex formed in the presence of ADP-BeF$_3$ may be disrupted and that the local map therefore represents an average of multiple orientations of the D1/D2 interface. The relative positions of the D1 and D2 domains are nevertheless consistent with our proposed model: The D2 domain is in the proper position to bind the alternate H90/H92 duplex region, and subsequent co-operative assembly of the D1/D2 interface unwinds the duplex and captures A-loop residues in the active site (Fig. 4m). This unwinding activity directly destabilizes the alternate structure of H90/H92 and favors the formation of the mature PTC secondary structure, since capture of A-loop residues in the Dbp10 active site does not interfere with the formation of the mature H90 and H92 stems.

### The CTT stabilizes the Dbp10 post-catalytic state

Our model of the Dbp10 post-catalytic state reflects the extensive RBF exchange that occurs after H90/H92 remodeling by Dbp10 (Fig. 2a, b). In this intermediate, Ssf1, Rrp14 and Rrp15 have been released from the pre-60S and replaced with a new set of RBFs; the Noc2/Noc3 heterodimeric complex bridges the L1 stalk and the newly docked rRNA domain III, stabilized by the extensive interaction network of the hub RBF Spb1[34,35] (Figs. 2e and 5a and Supplementary Fig. 1a). In this intermediate, the C-terminal portion of Dbp10 remains bound to the same pocket observed in the catalytic state, and in contrast with the catalytic state, the D1 domain is now clearly defined, allowing a complete atomic model of the catalytic domains of Dbp10 to be built (Fig. 5b). As expected, the H90 and H92 stems have their mature secondary structure and the compact tertiary junction anchoring H92 and the A-loop has also adopted its mature structure (Fig. 5c). This transition is consistent with our model, in which ATP hydrolysis and substrate release allow for the mature PTC structure to form (Fig. 5d).

Substrate release by Dbp10 does not lead to its disengagement from the pre-60S. Instead, three structural modules within the CTT interact with RBFs and rRNA elements to form a stable, non-catalytic assembly of the Dbp10 D1/D2 interface, in which the D1 domain is translated by ~10 Å and rotated by ~5° compared to the orientation of the catalytic intermediate (Fig. 6b). The first element (residues 808–834) runs across the subunit interface, directly stabilizing the non-catalytic D1/D2 interface and occupying the ssRNA binding pocket (Fig. 6b, c). The second element (residues 857–882) docks onto the D1 domain and mediates interactions with the upper segment of rRNA helix H61 (Fig. 6c). A third, more distal element (residues 918–946) runs along H64 and forms a specific interaction with the RBF Rlp24 (Fig. 6c). The H61/H64 junction, rather than being the substrate for Dbp10, is therefore bound post-catalytically. These C-terminal features are conserved in other post-catalysis structures of Dbp10/DDX54 (Supplementary Fig. 8a, b).

To understand the importance of the post-catalytic structure of Dbp10, we tested CTT deletions and alanine mutagenesis in the *DBP10* shuffle strain (Fig. 6a, d). Removal of the entire CTT tail resulted in a non-viable phenotype, even when a nucleolar targeting sequence (NoLS) was added to replace the predicted localization signal at the extreme C-terminus of Dbp10 (Fig. 6a, d). In contrast to the tail deletions, mutating or removing individual structural modules had no effect on growth (Fig. 6a, d), indicating that under normal growth conditions, disruption of individual contacts between the Dbp10 CTT and the pre-60S are not essential and suggesting a degree of redundancy between individual elements in stabilizing the post-catalytic Dbp10 intermediate.

### Helix H61 capture couples Dbp10 function to RBF exchange

Out of the six domains that make up the large ribosomal subunit, domain IV is the last one to become organized. In the pre-catalytic and catalytic states, the only ordered domain IV element is rRNA helix H61, which connects domain IV to the rest of the 25S rRNA. This conformation of H61, which is ~40 Å from the RNA binding pocket of Dbp10, is maintained by a conserved contact with the RBF Rrp14 (Fig. 6e). In the post-catalysis state, Rrp14 is no longer bound and the position of H61 pivots by ~60° about its anchoring point, allowing the H61/H64 junction to be engaged by Dbp10. Capture of the H61/H64 junction by Dbp10 disrupts the Rrp14/H61 interaction, weakening the affinity of Rrp14 for the pre-60S and promoting its dissociation (along with Ssf1 and Rrp15) from the pre-60S. Loss of Rrp14 exposes a core rRNA element formed by rRNA helices H25A and H32 which in the post-catalytic structure is occupied by the essential C-terminal module of Spb1 (Fig. 6f). Spb1 is a long, modular protein that acts as a hub protein to organize multiple RBFs during late nucleolar maturation[34,35]. Together, these sequential steps establish a chronology of how the remodeling function of Dbp10 is directly coupled to the changes in RBF composition during late nucleolar 60S maturation.

## Discussion

During RNP assembly, RNA folds in two phases: the initial co-transcriptional folding of stem-loop structures followed by the slower assembly of helical junctions and long-distance interactions to stabilize tertiary structures. During ribosome assembly, the latter is facilitated by the hierarchical incorporation of ribosomal proteins that stabilize these interactions. In bacteria, initial PTC formation appears to follow this pattern; stem-loops form their mature secondary structures prior to their full assembly into the PTC[36]. Surprisingly, our structural data reveal that, in eukaryotes, the universally conserved

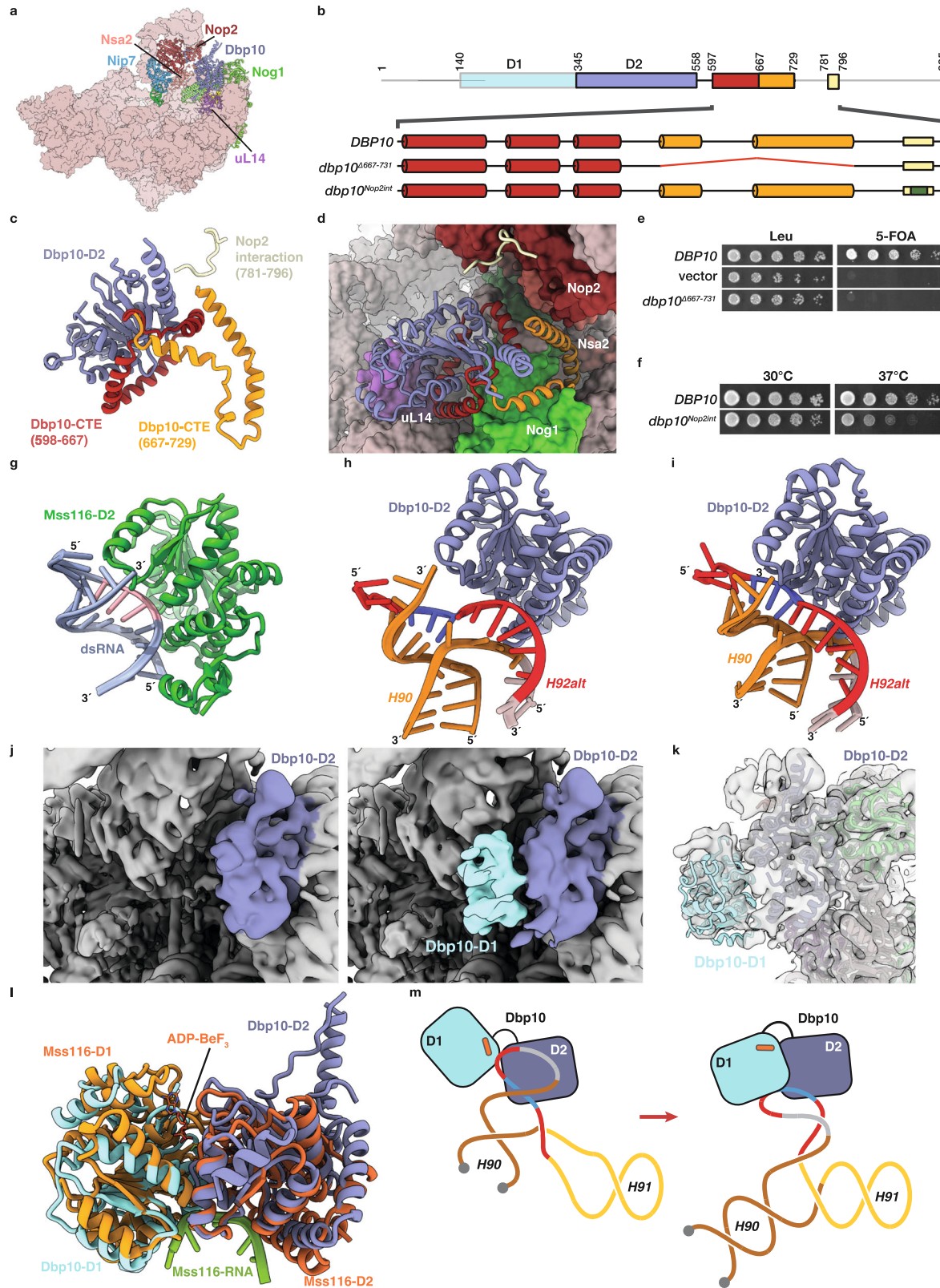

stem-loop H92 initially forms a different secondary structure and must be actively restructured before the tertiary structure of the PTC core can form. Restructuring is carried out by Dbp10/DDX54, which unwinds the immature H90/H92 duplex in an ATP-dependent manner to catalyze the formation of the mature PTC tertiary structure. One consequence of this mechanism is that additional sequence restraints are imposed on the PTC rRNA, to satisfy two sets of compatible base-

pairing interactions. Consistent with this, the degree of domain-specific conservation within the PTC is significantly higher in eukaryotes compared to archaea and bacteria[37,38]. A direct sequence comparison between *S.cerevisiae* and *H.sapiens* PTC sequences shows that all nucleotides required for the alternate H92 base pairing are identical (Supplementary Fig. 8c, d), implying a conserved function for Dbp10/DDX54 in eukaryotes. In contrast, Dbp10 function is distinct from the

**Fig. 4 | Structural analysis of catalytic Dbp10•pre-60S intermediates. a** Overall view of the Dbp10-bound 60S assembly intermediate, showing Dbp10 in the cleft formed by Nog1, Nsa2, Nop2 and uL14, colored and labeled as before. **b** Domain organization of Dbp10 highlighting two helical CTE elements (red and orange) and the Nop2-interaction motif (yellow) within the C-terminal tail (CTT). Positions of internal deletions (red) or alanine mutations (green) tested in panels (**e, f**) are also indicated. **c** Cartoon representation of D2 (blue), CTE elements (red and orange) and Nop2-interaction motif (yellow). **d** CTE-elements anchor Dbp10 to the pre-60S. Cartoon model of Dbp10 structural elements colored as in panel (**c**) with surface model of the 60S highlighting neighboring RBFs colored as in panel (**a**). Importance of CTE/CTT elements for Dbp10 function in vivo: (**e**) Internal deletion of the helical CTE finger element is lethal; (**f**) Alanine substitutions in the Nop2-interaction motif cause a temperature sensitive phenotype. **g** Cartoon model of the complex between Mss116 (green) and dsRNA (blue/pink) (PDB-4DB2[32]), capturing the initial dsRNA binding step. **h** Superposition of the H90/H92 pre-catalysis structure onto

the catalytic intermediate, showing positions of A-loop residues relative to D2. RNA is colored as in secondary structure diagrams in Fig. 3. **i** Speculative model of initial substrate engagement by Dbp10. The alternate H90/H92 was rigid-body shifted using Mss116/dsRNA as a guide. **j** Initial cryo-EM maps of the catalytic intermediate (low-pass filtered to 6 Å) prior to local classification, showing no defined density for D1. After two rounds of skip-align classification, D1 (cyan) has clear density. **k** Rigid body docking of D1 into locally refined maps low-pass filtered to 6 Å. **l** Comparison of the rigid-body refined Dbp10 D1/D2 domain (cyan and blue) to the structure of ssRNA-bound (green) Mss116 D1/D2 domains (yellow and orange – PDB-3I61[32]). **m** Schematic of the proposed mechanism for substrate engagement and unwinding by Dbp10. First, D2 engages the alternate duplex structure. D1 is bound to ATP (orange bar) but is initially uncoupled from D2. Subsequently, cooperative assembly of the D1/D2 interface unwinds the alternate duplex and initiates formation of mature H90/H92 base pairs. Source data are provided as a Source Data file.

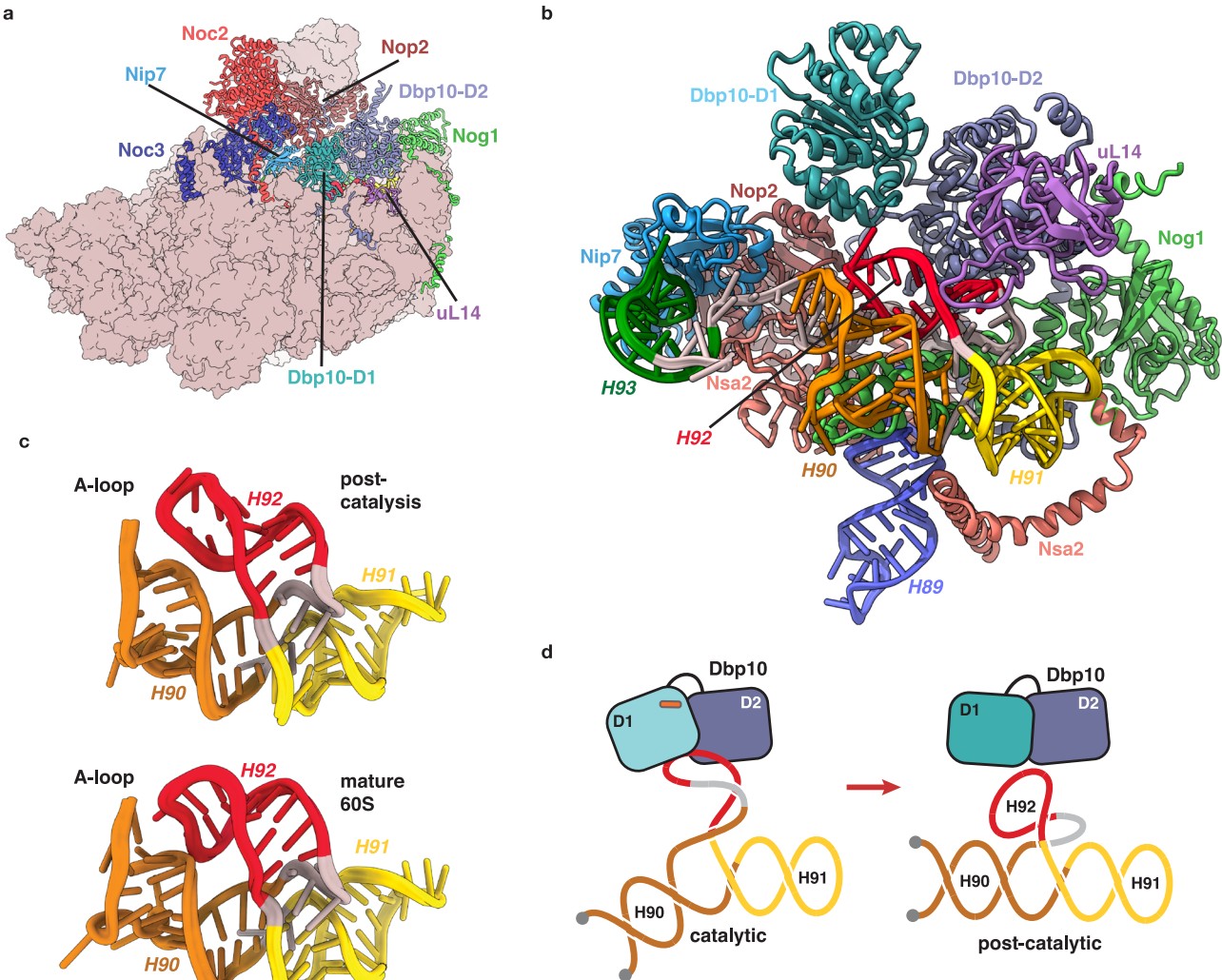

**Fig. 5 | The post-catalytic Dbp10·pre60S structure. a** Overview of the post-catalytic state, showing RBFs flanking Dbp10 as cartoon models, colored as before. **b** Detailed cartoon model of the post-catalytic state, showing the remodeled conformation of the H90/H92 helices below Dbp10. RBFs and rRNA are colored and labeled as before. **c** Comparison of post-catalytic (top) and mature (bottom – PDB-6TNU[9]) atomic models of helices H90/H91/H92 showing the Dbp10-mediated formation of the triple junction anchoring H92 and the A-loop. **d** Schematic of the

proposed mechanism for substrate release and H92 formation: first, engagement of A-loop residues by Dbp10 allows new base pairing interactions to initiate formation of H90 and H92 (left, identical to panel 4 m, right). Next, ATP hydrolysis and substrate release allows for assembly of the H90/H91/H92 junction and proper positioning of the A-loop. Dbp10 remains bound, but the D1/D2 interface is rotated into a non-catalytic state (right).

bacterial DEAD-box ATPase DbpA, which has a specific recognition domain that binds the mature form of the A-loop[39,40] and therefore does not appear to be involved in promoting H92 formation. The topology of the Dbp10 substrate is also important. The localized strain

imposed by the unwinding of the alternate conformation of H92 is only effective in inducing the restructuring of the alternate conformation because flanking PTC stem-loops are bound by RBFs. Local topology is also important for the function of the DEAD-box ATPase Spb4[34]. Our

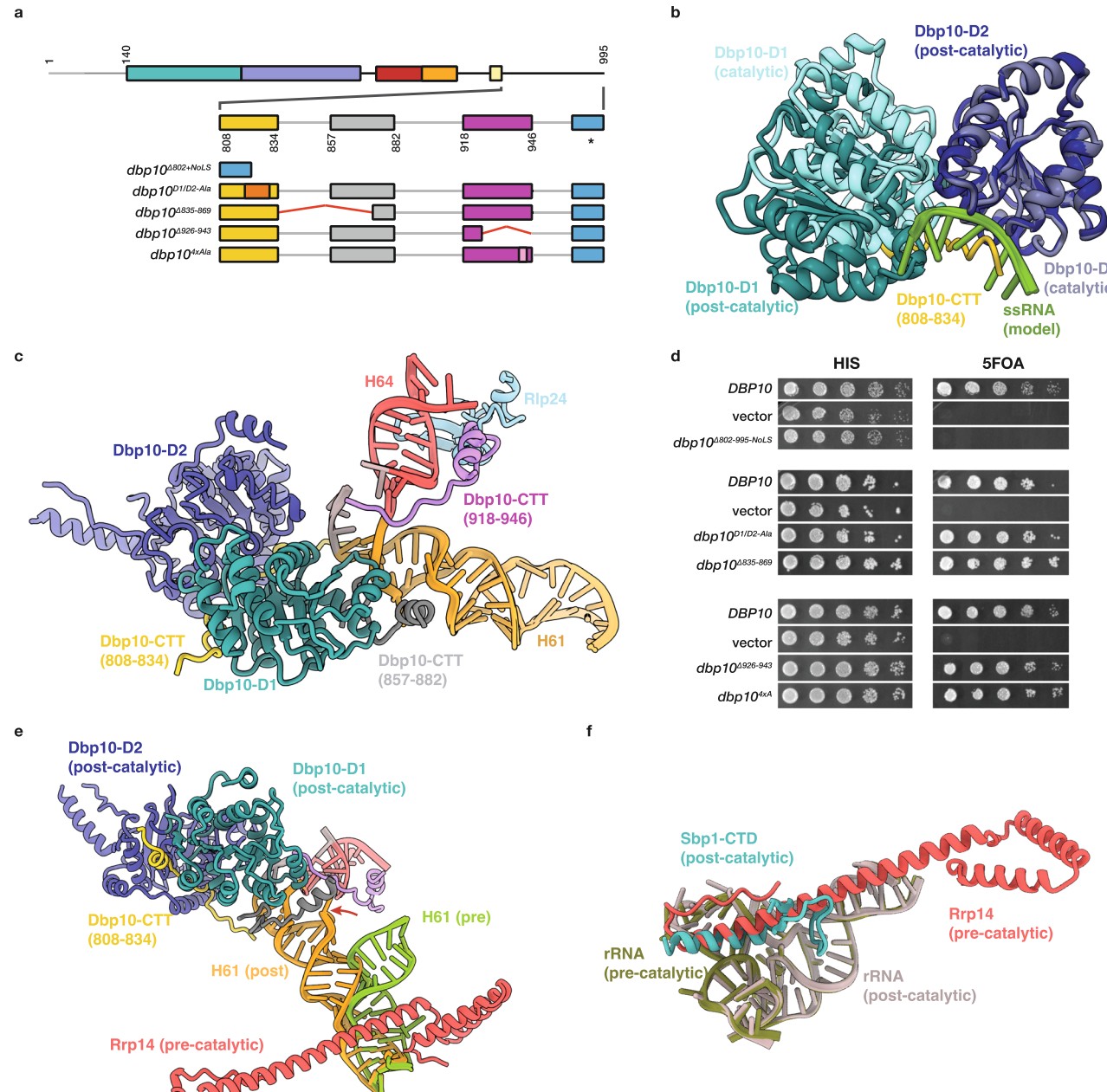

**Fig. 6 | Dbp10 promotes RBF exchange post-catalsis. a** Domain organization of Dbp10 highlighting regions of the Dbp10 C-terminal tail (CTT) that can be modeled in the post-catalytic maps. The D1/D2 bridging element is shown in dark yellow, the NTD-interacting region in gray and the H64-interaction module in magenta. The presumed nucleolar localization signal (NoLS) is shown in blue and marked with an asterisk. The positions of internal deletions (red lines) or alanine mutations (orange and pink) tested in panel (**d**) are indicated. **b** Cartoon representation of the post-catalytic orientation of D1/D2 (D1-dark blue, D2-dark teal) superposed on the closed D1/D2 model (cyan and blue) from the catalytic intermediate. The position of a model ssDNA substrate (green) is shown to highlight the fact that the D1/D2 bridge element of the CTT (yellow) occupies the ssRNA binding pocket in the post-catalysis state. **c** Cartoon representation of D1 (residues 857:882 - gray) and H64 (918:944 - magenta) interaction modules within the Dbp10 CTT, which stabilize the post-catalytic state by binding H61 and H64/Rlp24, respectively. **d** Importance of

the CTT in vivo. Deletion of the CTT is lethal (top panel), even when an orthologous NoLS is present. Alanine substitutions within the D1-D2 linker region or removal of structural elements engaging D1 are not essential for growth (middle panel). Deletion of the H64-interacting module or mutagenesis of residues mediating the interaction between the CTT and Rlp24 are also tolerated in vivo (bottom panel). **e** Post-catalytic repositioning of H61/H64 is incompatible with Rrp14/H61 binding. The interaction of Rrp14 (red) with the pre-catalytic 60S is partially mediated by interactions with H61 (olive). After PTC remodeling, Dbp10 (dark blue/teal) assumes its post-catalytic orientation and captures H61 (orange), weakening the Rrp14/H61 interaction. **f** The pre-60S binding sites of Rrp14 and the essential C-terminal element of Spb1 overlap. Because Dbp10 destabilizes the Rrp14/pre60S interaction in the post-catalytic state, it directly facilitates Spb1-CTD binding to promote RBF exchanges. Source data are provided as a Source Data file.

structures and proposed remodeling mechanism are therefore consistent with the localized duplex-unwinding model of DEAD-box ATPase function[41] and with the local topology constraints of the rRNA substrate[42]. Because Dbp10 is essential for H92 formation, our model also explains why Dbp10 function is necessary for A-loop

methylation by the RBF Sbp1, as only mature A-loops are recognized by this enzyme[24,34,35].

In addition to its essential catalytic role in PTC assembly, Dbp10 also plays a structural, post-catalytic role in guiding the ordered exchange of RBFs in late nucleolar 60S assembly. Upon substrate

release, the core domains of Dbp10 form a non-catalytic interface that captures the H61/H64 rRNA junction, a transition directly stabilized by several structural elements within the Dbp10 CTT. This transition communicates the successful completion of PTC assembly to more distal regions of the pre-60S by facilitating the release of Rrp14 and its interaction partners Ssf1 and Rrp15. In turn, Rrp14 release removes a steric block to allow the organization of the C-terminal anchoring domain of Spb1. In addition to its C-terminal methyltransferase domain, Spb1 contains multiple modules that interact with 15 other pre-60S components, including the Noc2/Noc3 heterodimer, acting as a critical hub to assemble rRNA domain III[19,20,30,34,35]. These sequential events therefore directly link PTC maturation and domain III organization, revealing how a non-equilibrium, ATP-dependent step is directly connected to subsequent structural changes in the pre-60S. After rRNA domain III is assembled, the pre-60S is acted on by another DEAD-box ATPase, Spb4, which catalyzes the organization of the H61/H62/H63/H64 junction, directly linking sequential ATP-dependent steps[34,35].

The recent publication of cryo-EM structures of human and *C. thermophilum* pre-60S intermediates bound by Dbp10/DDX54 in their post-catalytic configuration allow for a comparison of these steps across several eukaryotic species[19,20]. In both studies, intermediates were captured by pulling-down RBFs present throughout early and late assembly steps, capturing additional post-catalysis Dbp10/DDX54 intermediates that we did not observe with our *SPB4* depletion strategy and providing additional details on the sequential organization of rRNA domain IV. Despite the overall structural similarities across the two species, a comparison of these intermediates also suggests some variability in the order in which Rrp14/Rrp15/Ssf1 release and Noc2/Noc3/Spb1 binding occurs. This flexibility may also explain the high degree of functional redundancy observed in our mutational analysis of post-catalytic *S.cerevisiae* Dbp10 CTT elements. Our data and these studies are consistent with a model in which enzymatic steps such as those catalyzed by Dbp10/DDX54 require a highly defined substrate and are therefore highly conserved across all eukaryotes, while the order of non-energy consuming binding and release events is less rigidly constrained, allowing organism-specific variability.

The catalytic function of Dbp10 shows how limited local strand unwinding can trigger the remodeling of topologically restricted rRNA sequences. We had previously proposed a similar model for the DEAD-box ATPase Spb4, and it seems likely that at least a subset of the remaining 16 structurally uncharacterized DEAD-box ATPases involved in ribosome biogenesis function in a similar manner. Dbp10 and Spb4 both contribute to the formation of complex three-way rRNA junctions. Efficient formation of these rRNA junctions is important for two reasons. The first relates to the incredible demands placed on ribosome production in fast growing cells. For example, in yeast, approximately 2000 ribosomes per minute must be synthesized to maintain exponential growth, a rate requiring energy-consuming enzymes to accelerate kinetically slow RNA folding steps[14]. The second relates to the quality control mechanisms that ensure misfolded ribosomal intermediates are rapidly identified and recycled. While the precise details of this process are not understood, kinetic proof-reading has emerged as the leading hypothesis, proposing that failure to precisely execute rate-limiting steps during RNP biogenesis results in dead-end, degradation-prone intermediates[43,44]. This latter function may explain why DEAD-box ATPases are retained post-catalysis. Dbp10 is the third structurally characterized eukaryotic DEAD-box ATPase, after Has1 and Spb4, that remains associated with the pre-60S post-catalytically, suggesting that it is a common feature in eukaryotic ribosome assembly and may be a mechanism to communicate quality control during ribosome biogenesis. Finally, because Spb4 unwinds a segment of H63, the stabilization of the immature H61/H64 junction by Dbp10 opens the intriguing possibility that the post-catalysis structure of Dbp10 facilitates initial substrate binding by Spb4, thereby directly

coupling sequential DEAD-box protein unwinding events during ribosome biogenesis.

## Methods

### Yeast strains
Yeast strains used in this study are listed in Supplementary Table 1, plasmids used in this study are listed in Supplementary Table 2. Genomic knockout strains in yeast were performed using standard PCR protocols. For overexpression of *DBP10* variants, strains were grown overnight in YPD and then serially diluted onto either YPD or YPD + 2 µM β-estradiol and grown at 30 °C for 2 days. For shuffle assays, shuffle strains were transformed with the indicated plasmid, re-streaked onto selective media (SCD -LEU or SCD-HIS), and then grown overnight in selective media and serially diluted onto either selective media or 5-FOA and grown for 3 days at 30 °C.

### Plasmid construction
*DBP10* centromeric plasmids were generated as follows. A genomic fragment spanning 247 bp upstream *DBP10* to 66 bp downstream was cloned into pBluescript (Stratagene) using SacI/XhoI. The same sequence was cloned into pRS316 to generate pJE728 and pRS315 to generate pJE729. Point mutations and internal deletions were introduced into pJE729 using Quikchange (Agilent) to generate plasmids pJE748, pJE749, pJE750, pJE751, pJE753, pJE788, and pJE792. A synthesized DNA fragment (Genscript) was introduced carrying silent mutations in amino acids T188 and A189 to introduce an EagI site, V327 and D328 to introduce a SalI site, V565 and D566 to remove a SalI site, and amino acids L821, N822 and S823 to introduce an EcoRI site. The SacI/XhoI fragment was cloned into pRS313 using SacI and EcoRI to generate pJE1218. To generate pJE1222 (pRS313-*dbp10^Δnop2int*), a synthesized DNA fragment spanning BamHI to EcoRI was introduced to mutate the sequence from amino acids 787 to 794 from TFFLSHY to AAAASSA. To generate plasmid pJE1388, a synthesized DNA fragment was subcloned into pJE1218 deleting amino acids 926–943. To generate pJE1390, a synthesized DNA fragment was subcloned into pJE1322 mutating amino acids 931–934 from KHRD to AAAA. Plasmids pJE1384, pJE1386 and pJE1388 were generated using Quikchange and pJE1218 as a template. *DBP10* variants were introduced into overexpression plasmid pJE664 using XhoI/AscI.

### Microscopy
Strains were grown overnight in synthetic complete -uracil (SC-URA) medium with 2% dextrose, then diluted to an OD600 of 0.1 in SC-URA + 2% dextrose and grown to an OD of 0.4–0.5. Images were taken on a Nikon EclipseTi inverted microscope equipped 100× NA1.45 objective and an Andor Zyla 5.5 sCMOS camera. Image processing was performed using ImageJ.

### Polysome profiling
Cells were grown in YPD to mid-log phase (optical density at 600 nm (OD600) ≈ 0.8,) then treated with 50 µg/ml of cycloheximide for 5 min prior to centrifugation and snap-freezing. Thawed cells were washed and resuspended in lysis buffer (200 mM Tris-KCl, pH 7.4, 500 mM KCl, 100 mM MgCl₂, 1 mM DTT). Extracts were prepared by glass bead lysis (8 × 30 s) and clarified by centrifuging at 4 °C for 15 min at 11,300 × *g*. Then, 10 OD260 units of clarified extract were loaded onto 5–50% sucrose gradients made in the same buffer. Gradients were centrifuged for 2.5 h at 260,000 × *g* in a Beckman SW41 rotor and RNA content was monitored at 254 nm using a Gradient Master (BioComp).

### Overexpression and purification of Dbp10-containing pre-ribosomes
For yeast overexpression, full length wild-type or R522V catalytic mutant of *DBP10* was cloned into a modified pRS406 plasmid that

introduced a N-terminal 3xStrep-sfGFP-bdNEDD-MYC tag under the control of a β-estradiol inducible system. As a second affinity purification tag, *BRX1*, *SSF1* or *NOC2* was tagged at the genomic locus with a C-terminal 3xFLAG–3C–2xProteinA. Starter cultures for each strain were grown overnight in YPD to saturation. The following morning 12 L of YPD were inoculated from the starters to an OD600 of 0.1 and grown at 30 °C to an OD600 of 0.8. Overexpression was induced for 1 h by the addition of β-estradiol to a final concentration of 2 µM. Cells were harvested by centrifugation for 20 min at 4000 × *g*, the pellets were washed in Ribo-buffer A (50 mM Bis-Tris-Propane HCl pH 8.0, 125 mM NaCl, 25 mM KCl, 10 mM MgCl2, 1 mM TCEP and 0.1% (w/v) NP-40) and centrifuged again for 10 min at 4000 × *g*. Cell pellets were harvested then forced through a syringe into liquid nitrogen resulting in flash frozen yeast noodles. Cryogenic lysis was done using a grinding ball mill (Fritsch Pulverisette 6). For purification of pre-ribosomes, 30 g of lysate are warmed to 4 °C before addition of Ribo-buffer A supplemented with E64, pepstatin, PMSF and RNAse-free DNase I. Lysate was cleared by centrifuging at 100,000 × *g* for 30 min and loaded onto IgG Sepharose resin (Cytiva). Samples were washed with 10 column volumes of Ribo-buffer A followed by 10 column volumes of Ribo-buffer B (50 mM Bis-Tris-Propane HCl pH 8.0, 125 mM NaCl, 25 mM KCl, 10 mM MgCl2, 1 mM TCEP and 0.01% (w/v) NP-40). Protein-A tags were cleaved on-column for 1 h using 200 µg of 3C protease diluted into 3 mL of Ribo-buffer B and eluates were applied to a Strep-Tactin column (Cytiva), washed with 5 column volumes of Ribo-buffer B and also eluted by 30 min of on-column cleavage using 1 mL Ribo-buffer B supplemented with 50 µg of bdNEDD8 protease. The second elution for steady state sample derived from YJE1243 was instead performed with Ribo-buffer B supplemented with 10 mM desthiobiotin. In either case, eluted pre-ribosomes were concentrated on Amicon Ultra 0.5 ml spin columns with a 100KDa cutoff (Merck Millipore). Purification of post-catalysis Dbp10 which required depletion of SPB4 was performed by adding 3-Indole Acetic acid (IAA) to a final concentration of 0.5 mM at mid-log phase (OD600 of 0.8) for 45 min prior to harvesting, subsequent purification steps were identical to steady-state samples described above. For catalytic Dbp10•Ssf1 (steady state) and post-catalytic Dbp10•Noc2 (SPB4-AID) purification, all buffers were supplemented with 50 µM ADP-BeF3.

## Cryo-EM grid preparation and data collection

Cryo-EM grids were prepared by applying 3.5 µL of pre-ribosome sample at a concentration of 5 A260ml-1 to glow-discharged continuous carbon coated Quantifoil R2/1 300-mesh grids. Grids were blotted and frozen in liquid ethane using a Mark IV Vitrobot (FEI) set at 4 °C and 100% humidity. Micrographs were acquired on a Titan Krios (FEI) operated at 300KV equipped with a Gatan K3 direct electron detector using a slit width of 30 eV on a GIF-Quantum energy filter. Automated data collection was performed using SerialEM[45] with a defocus range of −0.9 to −2.2 µm and a pixel size of 1.08 Å. Each micrograph was dose fractionated into 30 frames of 0.05 s each under a dose rate of 26.2 e⁻/pixel/s with a total exposure time of 1.5 s and a dosage of approximately 39.3 e⁻/pixel. 10,879 movies were collected for the pre-catalytic sample from the inducible dominant-negative strain, 20,892 movies for the catalytic, steady state sample and 9452 movies for the post-catalytic sample from the SPB4-AID strain. The pre-catalytic sample was collected at the PNCC using a Titan Krios operated at 300KV with a pixel size of 1.0688 Å using the same parameters and defocus range as the other samples.

## Cryo-EM image processing

Motion correction was performed using MotionCorr2[46], and CTF parameters were estimated using GCTF[47]. All subsequent image processing was performed using RELION4[48].

## Pre-catalytic Dbp10[R522V]•Ssf1 dataset

About 300 particles were picked to generate initial 2D classes to serve as templates for automated particle picking from the 10,539 micrographs. 2734K particles were extracted, binned 4 times, and used for 2D classification, from these, 2374K particles were selected for 3D classification. All identical pre-60S classes were combined and re-extracted at the original pixel size of 1.08 Å, resulting in the total of 1863K particles. CTF and 3D refinement was performed with an imposed C1 symmetry and resulted in a reconstruction with a resolution of 2.66 Å without postprocessing. Three rounds of local skip-align 3D classification (T-factors of 60, 60 and 40) were performed using a mask surrounding the PTC region to subtract core density. From this, 375K particles were selected for the final reconstruction to an overall resolution of 2.52 Å. Additional local refinements were performed to improve the density around the L1 stalk, the PTC region and the Rrp14/Rrp15/Ssf1 region to facilitate model building in these regions. The final resolution in each case was estimated by applying a soft mask and was calculated using the gold-standard Fourier shell correlation (FSC) = 0.143[49]. Local resolution for all maps were generated using the ResMap[50] wrapper within Relion4.

## Catalytic Dbp10•Ssf1 (steady state) dataset

Initial 2D classes and processing was performed identically to the pre-catalytic dataset. Data from two separate grids (identical sample) were combined: in total, 3849K particles were extracted after automated particle picking from 20054 micrographs, binned 4 times and used for 2D classification. 2094K particles were selected for subsequent 3D classification. Identical classes were combined and re-extracted at the original pixel size of 1.08 Å, resulting in a combined total of 889K particles. 3D refinement was performed with an imposed symmetry of C1 and resulted in a reconstruction with a resolution of 2.9 Å before post-processing. Two rounds of local skip-align 3D classification (T-factors of 60 and 40) were performed using a mask surrounding the Dbp10 D1 region to subtract core density. From this, 195K particles were selected for the final reconstruction to an overall resolution of 2.7 Å. A local refinement of the domain VI region was also performed to improve the maps for model building and rigid body refinement. The final resolution in each case was estimated by applying a soft mask and was calculated using the gold-standard Fourier shell correlation (FSC) = 0.143[49]. Local resolution for all maps were generated using the ResMap[50] wrapper within Relion4.

## Post-catalytic Dbp10•Noc2 (SPB4-AID) dataset

Initial data processing was performed as described for the Dbp10•Ssf1 (steady-state) dataset. 1202K particles were selected from 8882 micrographs, binned 4 times and extracted for 2D classification. 869K particles were selected for 3D classification. Identical classes were combined resulting in 491K particles selected for further processing. Two rounds of local skip-align 3D classification (T-factors of 60 and 40) was performed using a mask surrounding the Dbp10/H64/H61region to subtract core density, yielding a final subset of 197K particles. After CTF refinement, polishing and postprocessing, a final map with a resolution of 2.66 Å was obtained. A local refinement of the domain VI region was also performed to improve the maps for model building. Two improve the maps surrounding the Dbp10-D1/H61/H64 region, another round of skip align classification (T-factor = 40) was performed with a more focused mask. Local refinement of this subset yielded improved maps of the H61/H64 region for model building.

## Cryo-EM model building and refinement

PDB 6EM1 was used as a starting model for the pre-catalytic Dbp10[R522V]•Ssf1 intermediates and the refined pre-catalytic model was used for building and refining into the catalytic Dbp10•Ssf1 (steady state) map. The C-terminal RecA domain of Dbp10, along

with the CTE and CTT were built de novo into this map. For the post-catalytic Dbp10•Noc2 (SPB4-AID) dataset, we used PDB 7R7A as a starting model and built the Dbp10 D1 domain using the Alphafold model[51]. All models were built in Coot[52] and refined in PHENIX using phenix.real_space_refine[53]. Figures were generated with UCSF Chimera X[54], movies were generated using PyMOL Molecular Graphics System (Version 2.0 Schödinger, LLC.).

## Purification of Brx1-Dbp10 containing pre-ribosomes

Starters of strains harboring a genomic C-terminal 3xFLAG-3C−2xProteinA tag on *BRX1* and either an inducible *DBP10* or *dbp10*[RS22V] were grown overnight in YPD to saturation. The following morning 12 L of YPD were inoculated from the starters to an OD600 of 0.2 and grown at 30 °C to an OD600 of 0.8 and overexpression was induced by adding β-estradiol to a final concentration of 2 μm. Cells were harvested and lysed identically to samples used for cryo-EM, described above. Lysate was cleared in the same way and loaded onto an IgG Sepharose resin (Cytiva). Samples were washed with 10 column volumes of Ribo-buffer A followed by 10 column volumes of Ribo-buffer B. Protein-A tags were cleaved on-column using 3C protease and eluates were directly loaded onto a Strep-Tactin column (Cytiva), washed with 5 column volumes of Ribo-buffer B and eluted by 30 min of on-column cleavage using 1 mL Ribo-buffer B supplemented with 50 μg of bdNEDD8 protease. Samples were concentrated to an A260ml⁻¹ of 3 (100 nM) using an Amicon Ultra 0.5 ml spin column with a 100KDa cutoff (Merck Millipore) to an OD260 absorbance of 5.0.

## Semi-quantitative mass-spectrometry analysis

10 μL of Dbp10 or Dbp10[RS22V] pre-ribosomes at an OD260 absorbance of 5.0 were run on a 4–20% denaturing Gel (BioRad) at 160 V for 7 min and stained with Coomassie Brilliant Blue R-250 (Thermo Fisher). The portion of the gel with protein bands was excised and used for subsequent analysis. Semi-quantitative mass-spectrometry data was obtained by the UTSW Proteomics core, analyzed using Proteome Discoverer 2.4 and searched against the yeast protein database from UniProt. Only high-confidence spectra (FDR ≤ 1%) were considered and are listed in Source Data. Selection of RBFs for our plot was based on their presence in known intermediate structures or well-established presences in specific pre-60S intermediates.

## Reporting summary

Further information on research design is available in the Nature Portfolio Reporting Summary linked to this article.

## Data availability

The cryo-EM density maps and models generated in this study have been deposited in the EMDB and PDB databases under accession codes: EMD-43017/PDB-8V83 (pre-catalytic structure – overall map and model), EMD-43018 (pre-catalytic structure – Local map PTC focused), EMD-43019 (pre-catalytic structure – Local map L1 focused), EMD-43020 (pre-catalytic structure – Local map Rrp14/Rrp15/Ssf1 focused), EMD-43021/PDB-8V84 (catalytic structure – overall map and model), EMD-43022 (catalytic structure – Local map Dbp10 focused), EMD-43024 (catalytic structure – Local map L1 focused), EMD-43026 (catalytic structure – Local map Rrp14/Rrp15/Ssf1 focused), EMD-43023/PDB-8V85 (catalytic structure – 6 Å low-pass-filtered, locally-refined map and docked D1 model), EMD-43027/PDB-8V87 (post-catalytic structure – overall map and model), EMD-43028 (post-catalytic structure – Local map Dbp10 focused), EMD-43029 (post-catalytic structure – Local map H61/H64 focused). Strain and plasmid information, cryo-EM micrographs, 2D classes, FSC curves, ResMap plots, and data processing 3D-classification and particle sorting schemes generated in this study are provided in the Supplementary Information. Source data are provided with this paper.

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

## Acknowledgements

We thank Sahana Balasubramanian and Rafal Piwowarczyk for generating strains and plasmids during the early stages of this project. We also would like to thank the UTSW core facilities that have supported this work: Daniel Stoddard and the staff the UTSW Cryo-Electron Microscopy Facility, funded in part by the CPRIT Core Facility Support Award RP170644, James Chen at the UTSW Structural Biology Lab, Andrew Lemoff at the UTSW Proteomics Core and Marcel Mettlen at the UTSW Quantitative Light Microscopy Core, a shared resource of the Harold C. Simmons Cancer Center, supported in part by an NCI Cancer Center Support Grant, 1P30 CA142543-01. A portion of this research was supported by NIH grant U24GM129547 and performed at the PNCC at OHSU and accessed through EMSL (grid.436923.9), a DOE Office of Science User Facility sponsored by the Office of Biological and Environmental Research, with the assistance of Theo Humphreys. J.P.E. was supported by the Cancer Prevention and Research Institute of Texas (RR150074), the Welch Foundation (I-1897), the UTSW Endowed Scholars Fund and the National Institutes of Health (GM135617-01).

## Author contributions

V.E.C., C.S.W, N.P and J.P.E. conceived the project. V.E.C., C.S.W., and N.P. constructed strains. V.E.C., and N.P. carried out sample preparations and collected cryo-EM data. C.S.W. conducted polysome profiling, growth, and microscopy assays. V.E.C., and J.P.E. processed the cryo-EM data. The paper was written by J.P.E., V.E.C and C.S.W and edited by V.E.C., C.S.W. and J.P.E with input from N.P. J.P.E. supervised the work.

## Competing interests

The authors declare no competing interests.
