## [Peer Review File · Nature Communications]

The DEAD-box ATPase Dbp10/DDX54 initiates peptidyl transferase center formation during 60S ribosome biogenesisREVIEWER COMMENTS

Reviewer #1 (Remarks to the Author):

The flurry of cryo-EM structures of 60S pre-ribosomes in different states is a challenge for the ribosome synthesis community. Erzberger and co-workers report new 60S pre-ribosome structures and focus particularly on the molecular environment of the RNA helicase Dbp10. The structures and discussions provide clarity and insights into the role of this conserved RNA helicase during late nucleolar assembly steps.

Specifically, it provides important insights into what is happening before and after Dbp10 action on the 60S pre-ribosome. The structures per se do not add a dramatic amount of new information, since they are very similar to what has been published (Lau et al). The contacts of Dbp10 with the rRNA and with other AFs are very similar. Although, Cruz et al suggest that the structures of Dbp10 in the Lau paper are post-catalytic, so they add a pre-catalytic state. Cruz et al focus on Dbp10 and its immediate effects on the surrounding AF, such as Sbp4, Spb1, Noc2/Noc3 or Rrp14/Rrp15/Ssf1. The paper adds nice information about the release of the Ssf1 module and the role of Rrp14. The paper discusses extensively about the need for Dbp10 to work on H90-92 and how this function is important to form the A-loop of the PTC. In contrast, in the Lau et al paper, they emphasize how Dbp10 keeps H61/64 open and avoid premature contacts, suggesting that Dbp10 safeguards the initial development of domain IV. Cruz et al also describe about H61/64 contacts, and claim that these RNA helices are not the direct Dbp10 substrate, but are bound post-catalytically, which is interesting and provides a more complete picture of PTC development.

Overall, the manuscript is well written and the structures have been carefully analysed and thought through; and the work is put in context of the nucleolar maturation pathway of the 60S subunit. I do not have any suggestions or critique to improve the current version, and therefore support publication.

Reviewer #2 (Remarks to the Author):

Cruz et al present a series of cryo-EM structures of the pre-60S revealing new insight into the formation of the peptidyl transferase center (PTC). Dbp10 is a DEAD-box helicase required for the maturation of the large ribosomal subunit but its precise catalytic function has remained undefined. To probe the function of Dbp10 the authors designed a series of mutants to block ATP binding, RNA binding, or ATP hydrolysis. Using a DBP10 shuffle strain the authors demonstrate that all of these mutants cause either a slow growth or lethal phenotype and impair ribosome assembly as evidenced by sucrose gradients and nuclear accumulation of pre-60S particles. Mass spec of pre-60S particles generated with the R522V mutant revealed that Dbp10 remains bound to the pre-60S following ATP hydrolysis – similar to other ribosome assembly helicases.

Next the authors used a combination of genetic and biochemical approaches to trap pre-60S particles at different stages in the Dbp10 catalytic cycle. Three cryo-EM structures were determined including a pre-catalytic state without Dbp10 (protein dissociates during sample prep), pre-catalytic state with Dbp10 and BeF₃, and a Dbp10 bound post-catalytic state. Both pre-catalytic structures (with and without Dbp10) revealed that the PTC is in an immature state with the H92 stem having an alternative base-pairing arrangement to what is seen in the mature PTC. The discovery of the immature A-loop is noteworthy, as this is distinct from PTC formation in bacteria. The pre-catalytic cryo-EM structures suggest that Dbp10 functions to unwind this alternative state to promote the formation of the mature PTC. This is also supported by modeling using the coordinates of the related helicase Mss116. Finally, the structure of the post-catalytic Dbp10 structure revealed the large exchange of assembly factors that occurs following remodeling of the PTC. The significance of several Dbp10 interfaces was confirmed in yeast using the Dbp10 shuffle strain. Overall, this is a comprehensive body of work supported by both cryo-EM and genetics that reveals very significant new details about the formation of the PTC A-loop, during ribosome assembly.

I fully support publication of this manuscript, but I have several suggestions for improvement:

1. Figure 1A – Where do these Dbp10 coordinates come from? Please provide more details in the figure legend and the appropriate PDB code(s).

2. Figure 3 – The secondary structure diagrams are very helpful to understand the difference between the immature and mature states of the PTC. It would be useful to readers to add labels on the figure for the immature (panels f and g) and mature (panels h and i) states.

3. It is important to show readers the local resolution for Dbp10 in both the pre and post-catalytic structures. While this information is shown in extended data figures, only the EM density is shown thus it is impossible to figure out what is what. The authors should remake these figures and clearly label the domains of Dbp10. Showing the local resolution surrounding H92alt is also important.

4. Figure 4 – I have several suggestions for this figure:

a. Several of the panels should be moved to a new supplemental figure so that the font size can be increased. For example, I can't read the text in panel b and e.

b. The colors used for the different domains of Dbp10 are confusing with the colors used for H90 and H92alt as well as the colors used for Mss116. Please use something other than orange and red.

c. It would be useful to add the labels D1 and D2 to the boxes in panel b.

d. Please define the orange line in the D1 domain in panel m.

5. Figure 5 – Similar comments to figure 4

a. The text in panels e and h is too small to read. Several parts of this figure could be moved to a supplemental figure.

b. In panel d change the color of the D1 and D2 domain in the post-catalytic state to reflect the other panels. The same colors should also be used in panel a and b.

6. In the conclusion the authors discuss similarities and differences between recent Dbp10/DDX54 bound pre-60S structures from *C. thermophilum* and human. It would be helpful to show a side-by-side comparison of these in a supplemental figure. Are there any significant differences between Dbp10 homologues? Are all the domains/regions in the C-terminus conserved?

7. While this manuscript was under review a new paper was published by the Hurt lab that further supports the role of Dbp10 in PTC formation (Mitterer et al NAR, 2023). The authors should incorporate this new information into the discussion/conclusion.

Reviewer #3 (Remarks to the Author):

The assembly of ribosomes requires the precise folding and modification of the ribosomal RNAs. In eukaryotic cells, several DEAD box ATPases are essential for the production of ribosomes, presumably for RNA remodeling. While extensive molecular genetic and biochemical analysis has given us some understanding of when these enzymes function in the pathway and what protein exchanges they promote, a detailed mechanistic understanding of their specific functions in RNA remodeling requires structural analysis. The catalytic center of the ribosome, the peptidyl transferase center, properly positions peptidyl and amino acyl tRNAs for chemistry. In particular, the A loop of H92 stabilizes the incoming amino acyl tRNA. In this manuscript, the authors determine structures for pre-60S complexes associated with Dbp10 pre and post ATP hydrolysis. They also attempted to obtain the structure of a true catalytic intermediate using ADP-BeF₃. Unfortunately, in the catalytic intermediate the RNA substrate was not stably engaged with the ATPase. Nevertheless, by comparing the pre and post catalytic structures, the authors provide compelling evidence that Dbp10 remodels the A loop into its mature conformation. This is an interesting result because in bacteria, the A loop adopts its mature conformation upon initial co-transcriptional folding of secondary structure and does not require a remodeling activity such as Dbp10. Altogether the work appears solid and the conclusions warranted, however there are many instances where the presentation can be made clearer to improve its readability.

Major points:

1. The authors describe characterizing the catalytic intermediate in two sections. This redundancy is confusing and this portion of the manuscript should be rewritten. (P4L16) “Therefore, we set out to isolate substrate-bound Dbp10-60S intermediates.... The RBF composition of this intermediate is indistinguishable from the pre-catalysis state, except for extra density corresponding to the C-terminal portion of Dbp10” But this was already described (P2L20) “A 2.7Å reconstruction of the catalytic Dbp10 intermediate was obtained. ... This structure is compositionally identical to the pre-catalytic state, except for the presence of Dbp10”

2. In the section “Dbp10 unwinding activity destabilizes the immature A-loop” the authors must comment on the status of the substrate RNA. They later acknowledge that the RNA was not observed but this needs to be stated up front to provide the rationale for the superposition comparison that they provide.

3. It would strengthen the author's case that in eukaryotes in general and not just yeast that the A loop adopts two different secondary structure if they could show that the alt conformation of H92 is supported by predicted conservation of secondary structure compatible with the two folded states of H92.

Minor points:

1. It doesn't make sense to say that Dbp10 was stoichiometric in the purified particles but washed off during purification: “Purification of pre-60S particles ... yielded purified particles with stoichiometric amounts of Dbp10. However, ...Dbp10, ... dissociated during purification.”

2. Clarify – a subset of particles “we focused on the predicted binding region of Dbp10 and selected a subset with a distinct.”

3. In several places, the authors refer to the L1 stalk but it is not indicated in the figures and there is no context to understand reference to the L1 stalk.

4. In several places the authors refer to the RNA domains of the large subunit. For example, “Out of the six domains that make up the large ribosomal subunit, domain IV is...” A figure showing the different RNA domains would give these references context.

5. In the section “The CTT stabilizes the Dbp10 post-catalytic state” the authors refer to Spb1 as an interaction hub and reference Figs 2e and 5a. There is no panel 2e and it is not evident from their figures that Spb1 is a hub.

6. The section “Dbp10 rRNA remodeling triggers an extensive rearrangement of the pre-60S” would benefit from a cartoon illustrating these rearrangements.

7. In Fig 1F, Tif6 is the most reduced in abundance, greater than Noc2 and Noc3. However, Tif6 is present in all the reconstructed particles. In addition, the absolute changes in abundances < 50% are less than expected. Why is this?

8. In Fig 3, additional labels in the figure would help the reader. For example, if f and g represent pre-catalytic and h and i represent post-catalytic states, this should be indicated in the figure itself.

9. In Fig 4 h, l and M identify that the gray and blue segments of RNA represent.

10. Fig 4m is confusing. The cartoon on the right is described first and described as the pre-catalytic structure “(Right) engagement of the alternate duplex.” The legend suggests progression from right to left but the figure clearly indicates the transition is from the left to right. Indicate when nucleotide is bound and hydrolyzed. Clarify what are the gray and blue elements of the RNA and where the A loop is. In Fig 2g and i use the same color coding - blue and gray.

11. It is not clear how the cartoon in Fig 4m relates to that in Fig 5d. Clarify what different aspects of Dbp10 activity these cartoons show.

Reviewer #1 (Remarks to the Author):

The flurry of cryo-EM structures of 60S pre-ribosomes in different states is a challenge for the ribosome synthesis community. Erzberger and co-workers report new 60S pre-ribosome structures and focus particularly on the molecular environment of the RNA helicase Dbp10. The structures and discussions provide clarity and insights into the role of this conserved RNA helicase during late nucleolar assembly steps.

Specifically, it provides important insights into what is happening before and after Dbp10 action on the 60S pre-ribosome. The structures per se do not add a dramatic amount of new information, since they are very similar to what has been published (Lau et al). The contacts of Dbp10 with the rRNA and with other AFs are very similar. Although, Cruz et al suggest that the structures of Dbp10 in the Lau paper are post-catalytic, so they add a pre-catalytic state. Cruz et al focus on Dbp10 and its immediate effects on the surrounding AF, such as Sbp4, Spb1, Noc2/Noc3 or Rrp14/Rrp15/Ssf1. The paper adds nice information about the release of the Ssf1 module and the role of Rrp14. The paper discusses extensively about the need for Dbp10 to work on H90-92 and how this function is important to form the A-loop of the PTC. In contrast, in the Lau et al paper, they emphasize how Dbp10 keeps H61/64 open and avoid premature contacts, suggesting that Dbp10 safeguards the initial development of domain IV. Cruz et al also describe about H61/64 contacts, and claim that these RNA helices are not the direct Dbp10 substrate, but are bound post-catalytically, which is interesting and provides a more complete picture of PTC development. Overall, the manuscript is well written and the structures have been carefully analysed and thought through; and the work is put in context of the nucleolar maturation pathway of the 60S subunit. I do not have any suggestions or critique to improve the current version, and therefore support publication.

We thank the reviewer for his comments and for supporting publication of the manuscript.

Reviewer #2 (Remarks to the Author):

Cruz et al present a series of cryo-EM structures of the pre-60S revealing new insight into the formation of the peptidyl transferase center (PTC). Dbp10 is a DEAD-box helicase required for the maturation of the large ribosomal subunit but its precise catalytic function has remained undefined. To probe the function of Dbp10 the authors designed a series of mutants to block ATP binding, RNA binding, or ATP hydrolysis. Using a DBP10 shuffle strain the authors demonstrate that all of these mutants cause either a slow growth or lethal phenotype and impair ribosome assembly as evidenced by sucrose gradients and nuclear accumulation of pre-60S particles. Mass spec of pre-60S particles generated with the R522V mutant revealed that Dbp10 remains bound to the pre-60S following ATP hydrolysis – similar to other ribosome assembly helicases.

Next the authors used a combination of genetic and biochemical approaches to trap pre-60S particles at different stages in the Dbp10 catalytic cycle. Three cryo-EM structures were determined including a pre-catalytic state without Dbp10 (protein dissociates during sample prep), pre-catalytic state with Dbp10 and BeF₃, and a Dbp10 bound post-catalytic state. Both pre-catalytic structures (with and without Dbp10) revealed that the PTC is in an immature state with the H92 stem having an alternative base-pairing arrangement to what is seen in the mature PTC. The discovery of the immature A-loop is noteworthy, as this is distinct from PTC formation in bacteria. The pre-catalytic cryo-EM structures suggest that Dbp10 functions to unwind this alternative state to promote the

formation of the mature PTC. This is also supported by modeling using the coordinates of the related helicase Mss116. Finally, the structure of the post-catalytic Dbp10 structure revealed the large exchange of assembly factors that occurs following remodeling of the PTC. The significance of several Dbp10 interfaces was confirmed in yeast using the Dbp10 shuffle strain. Overall, this is a comprehensive body of work supported by both cryo-EM and genetics that reveals very significant new details about the formation of the PTC A-loop, during ribosome assembly.

I fully support publication of this manuscript, but I have several suggestions for improvement:

We thank the reviewer for his comments and for supporting publication of the manuscript.

1. Figure 1A – Where do these Dbp10 coordinates come from? Please provide more details in the figure legend and the appropriate PDB code(s).

The figure legend has been amended to clarify that this is just a schematic of universally conserved DEAD-box ATPase active site residues and not coordinates from a structure solved in this paper.

2. Figure 3 – The secondary structure diagrams are very helpful to understand the difference between the immature and mature states of the PTC. It would be useful to readers to add labels on the figure for the immature (panels f and g) and mature (panels h and i) states.

These labels have been added to the figure.

3. It is important to show readers the local resolution for Dbp10 in both the pre and post-catalytic structures. While this information is shown in extended data figures, only the EM density is shown thus it is impossible to figure out what is what. The authors should remake these figures and clearly label the domains of Dbp10. Showing the local resolution surrounding H92alt is also important.

We added a new panel to supplemental figures 3, 5 and 7 that shows a detailed view of the local resolution near H92alt and Dbp10 in both pre- and post-catalytic states. The relevant densities have been labeled in each panel.

4. Figure 4 – I have several suggestions for this figure:

a. Several of the panels should be moved to a new supplemental figure so that the font size can be increased. For example, I can't read the text in panel b and e.

The font sizes in panels b and e were increased to improve readability.

b. The colors used for the different domains of Dbp10 are confusing with the colors used for H90 and H92alt as well as the colors used for Mss116. Please use something other than orange and red.

We agree that the using a different color palette for the Mss116 structure would make the figure less confusing and have altered panel g in Figure 4.

c. It would be useful to add the labels D1 and D2 to the boxes in panel b.

d. Please define the orange line in the D1 domain in panel m.

D1 and D2 are now labeled in panel b. The orange bar in panel m has been defined as ATP in the figure legend.

5. Figure 5 – Similar comments to figure 4

a. The text in panels e and h is too small to read. Several parts of this figure could be moved to a supplemental figure.

b. In panel d change the color of the D1 and D2 domain in the post-catalytic state to reflect the other panels. The same colors should also be used in panel a and b.

We have increased the font size in all panels and have replaced panels a and b with versions that individually color the two domains of Dbp10.

6. In the conclusion the authors discuss similarities and differences between recent Dbp10/DDX54 bound pre-60S structures from *C. thermophilum* and human. It would be helpful to show a side-by-side comparison of these in a supplemental figure. Are there any significant differences between Dbp10 homologues? Are all the domains/regions in the C-terminus conserved?

We have made additional figure panels (supplementary figure 8a,b) that compares the available Dbp10/DDX54 molecular structures and added the following description the section describing the post-catalysis structures: “These C-terminal tail features are conserved in other post-catalysis structures of Dbp10/DDX54 (Supplementary Fig. 8a,b).”

7. While this manuscript was under review a new paper was published by the Hurt lab that further supports the role of Dbp10 in PTC formation (Mitterer et al NAR, 2023). The authors should incorporate this new information into the discussion/conclusion.

We have added the following passage to the results section: “Our results are broadly consistent with a study of Dbp10 mutants published while this manuscript was under review.” and this passage to the discussion section: “Because Dbp10 is essential for H92 formation, our model also explains why Dbp10 function is necessary for A-loop methylation by the RBF Sbp1, as only mature A-loops are recognized by this enzyme”. The Mitterer paper is cited in both instances.

Reviewer #3 (Remarks to the Author):

The assembly of ribosomes requires the precise folding and modification of the ribosomal RNAs. In eukaryotic cells, several DEAD box ATPases are essential for the production of ribosomes, presumably for RNA remodeling. While extensive molecular genetic and biochemical analysis has given us some understanding of when these enzymes function in the pathway and what protein exchanges they promote, a detailed mechanistic understanding of their specific functions in RNA remodeling requires structural analysis. The catalytic center of the ribosome, the peptidyl transferase center, properly positions peptidyl and amino acyl tRNAs for chemistry. In particular, the A loop of H92 stabilizes the incoming amino acyl tRNA. In this manuscript, the authors determine structures for pre-60S complexes associated with Dbp10 pre and post ATP hydrolysis. They also attempted to obtain the structure of a true catalytic intermediate using ADP-BeF₃. Unfortunately, in the catalytic intermediate the RNA substrate was not stably engaged with the ATPase. Nevertheless, by comparing the pre and post catalytic structures, the authors provide compelling evidence that

Dbp10 remodels the A loop into its mature conformation. This is an interesting result because in bacteria, the A loop adopts its mature conformation upon initial co-transcriptional folding of secondary structure and does not require a remodeling activity such as Dbp10. Altogether the work appears solid and the conclusions warranted, however there are many instances where the presentation can be made clearer to improve its readability.

We thank the reviewer for his comments and suggestions.

Major points:

1. The authors describe characterizing the catalytic intermediate in two sections. This redundancy is confusing and this portion of the manuscript should be rewritten. (P4L16) “Therefore, we set out to isolate substrate-bound Dbp10-60S intermediates.... The RBF composition of this intermediate is indistinguishable from the pre-catalysis state, except for extra density corresponding to the C-terminal portion of Dbp10” But this was already described (P2L20) “A 2.7Å reconstruction of the catalytic Dbp10 intermediate was obtained. ... This structure is compositionally identical to the pre-catalytic state, except for the presence of Dbp10”

The second reference to the rationale for this data collection set has been deleted.

2. In the section “Dbp10 unwinding activity destabilizes the immature A-loop” the authors must comment on the status of the substrate RNA. They later acknowledge that the RNA was not observed but this needs to be stated up front to provide the rationale for the superposition comparison that they provide.

We expanded on this section of the manuscript: “The absence of defined substrate ssRNA at the D1/D2 interface could be due either to the presence of multiple states representing intermediates between H92 alt and the mature PTC conformation or because of destabilization of the complex during grid preparation. In particular, because the DSS crosslinker added to stabilize our intermediate can readily form monolinks with lysine residues, we speculate that the Dbp10/ssRNA complex formed in the presence of ADP-BeF₃ may be disrupted and that the local map therefore represents an average of multiple orientations of the D1/D2 interface.”

3. It would strengthen the author's case that in eukaryotes in general and not just yeast that the A loop adopts two different secondary structure if they could show that the alt conformation of H92 is supported by predicted conservation of secondary structure compatible with the two folded states of H92.

We have added two panels to a new supplementary figure (S8c,d) which show that all residues involved in H92alt formation are identical in *S.cerevisiae* and *H.sapiens* and have added the following sentence to the manuscript: “A direct sequence comparison between *S.cerevisiae* and *H.sapiens* PTC sequences shows that all nucleotides required for the alternate H92 base pairing are identical, implying a conserved function for Dbp10/DDX54 in eukaryotes.”

Minor points:

1. It doesn't make sense to say that Dbp10 was stoichiometric in the purified particles but washed

off during purification: “Purification of pre-60S particles ... yielded purified particles with stoichiometric amounts of Dbp10. However, ...Dbp10, ... dissociated during purification.”

This sentence has been clarified to read “However, cryo-EM reconstructions of this sample yielded a reconstruction that lacked any density for Dbp10, likely because it dissociated during grid preparation.”

2. Clarify – a subset of particles “we focused on the predicted binding region of Dbp10 and selected a subset with a distinct.”

We expanded this sentence to clarify our reasoning: ”A skip align 3D-classification focused on the region predicted by XL-MS to represent the binding region of Dbp10²⁶ revealed a subset of particles with a distinct, previously uncharacterized structural feature. Because Dbp10^{R522V} is catalytically impaired, we reasoned that the resulting 2.5 Å reconstruction might represent the pre-catalytic state.”

3. In several places, the authors refer to the L1 stalk but it is not indicated in the figures and there is no context to understand reference to the L1 stalk.

The L1 stalk has been defined on page 4 as “but may instead help stabilize the mobile RNA segment termed the L1 stalk (helices H75-H78), especially at higher temperatures.”

4. In several places the authors refer to the RNA domains of the large subunit. For example, “Out of the six domains that make up the large ribosomal subunit, domain IV is...” A figure showing the different RNA domains would give these references context.

We added a panel to Supplementary Figure 1 that shows a schematic of the rRNA domains.

5. In the section “The CTT stabilizes the Dbp10 post-catalytic state” the authors refer to Spb1 as an interaction hub and reference Figs 2e and 5a. There is no panel 2e and it is not evident from their figures that Spb1 is a hub.

This has been corrected to Fig. 2d. The hub function of Spb1 has been extensively characterized. We have added the appropriate references.

6. The section “Dbp10 rRNA remodeling triggers an extensive rearrangement of the pre-60S” would benefit from a cartoon illustrating these rearrangements.

We have created a supplemental movie that illustrates the structural rearrangements that connect the catalytic and post-catalytic roles of Dbp10 and how H92alt remodeling is directly connected to the exchange of RBFs during nucleolar 60S maturation.

7. In Fig 1F, Tif6 is the most reduced in abundance, greater than Noc2 and Noc3. However, Tif6 is present in all the reconstructed particles. In addition, the absolute changes in abundances < 50% are less than expected. Why is this?

The reviewer correctly points out that the differences in Tif6 abundance between the two strains is puzzling. Certain outliers are expected in this type of analysis and our

interpretation relies on the broader, overall pattern rather than the changes in individual components that may represent outliers. Because Dbp10^{R522V} retains some catalytic activity, intermediate accumulation is expected to be less severe than one resulting from complete disruption of Dbp10 function.

8. In Fig 3, additional labels in the figure would help the reader. For example, if f and g represent pre-catalytic and h and i represent post-catalytic states, this should be indicated in the figure itself.

The figure has been amended so that panels f/g are labeled “immature-PTC” and h/i are “mature-PTC.”

9. In Fig 4 h, I and M identify that the gray and blue segments of RNA represent.

We have amended the figure legend to explain that the coloring mirrors that of the secondary structure diagram in Figure 3.

10. Fig 4m is confusing. The cartoon on the right is described first and described as the pre-catalytic structure “(Right) engagement of the alternate duplex.” The legend suggests progression from right to left but the figure clearly indicates the transition is from the left to right. Indicate when nucleotide is bound and hydrolyzed. Clarify what are the gray and blue elements of the RNA and where the A loop is. In Fig 2g and i use the same color coding - blue and gray.

The correct transition is from left to right. The figure legend has been clarified to read, “Schematic of the proposed mechanism for substrate engagement and unwinding by Dbp10. First, the D2 domain engages the alternate duplex structure. D1 is bound to ATP (orange bar) but is initially uncoupled from D2. Subsequently, cooperative assembly of the D1/D2 interface unwinds the alternate duplex and initiates the formation of mature H90/H92 base pairs.”

11. It is not clear how the cartoon in Fig 4m relates to that in Fig 5d. Clarify what different aspects of Dbp10 activity these cartoons show.

Figure legend has been amended to read, “Schematic of the proposed mechanism for substrate engagement and unwinding by Dbp10. First, the D2 domain engages the alternate duplex structure. D1 is bound to ATP (orange bar) but is initially uncoupled from D2. Subsequently, cooperative assembly of the D1/D2 interface unwinds the alternate duplex and initiates the formation of mature H90/H92 base pairs.”